## Housing, neighbourhood and sociodemographic associations with adult levels of physical activity and adiposity: baseline findings from the ENABLE London study

Claire M Nightingale,[1] Alicja R Rudnicka,[1] Bina Ram,[1] Aparna Shankar,[1] Elizabeth S Limb,[1] Duncan Procter,[2,3] Ashley R Cooper,[2,3] Angie S Page,[2,3] Anne Ellaway,[4] Billie Giles-Corti,[5] Christelle Clary,[6] Daniel Lewis,[6] Steven Cummins,[6] Peter H Whincup,[1] Derek G Cook,[1] Christopher G Owen[1]

For numbered affiliations see end of article.

**Correspondence to**
Dr Claire M Nightingale;
cnightin@sgul.ac.uk

## ABSTRACT

**Objectives** The neighbourhood environment is increasingly shown to be an important correlate of health. We assessed associations between housing tenure, neighbourhood perceptions, sociodemographic factors and levels of physical activity (PA) and adiposity among adults seeking housing in East Village (formerly London 2012 Olympic/Paralympic Games Athletes' Village).

**Setting** Cross-sectional analysis of adults seeking social, intermediate and market-rent housing in East Village.

**Participants** 1278 participants took part in the study (58% female). Complete data on adiposity (body mass index (BMI) and fat mass %) were available for 1240 participants (97%); of these, a subset of 1107 participants (89%) met the inclusion criteria for analyses of accelerometer-based measurements of PA. We examined associations between housing sector sought, neighbourhood perceptions (covariates) and PA and adiposity (dependent variables) adjusted for household clustering, sex, age group, ethnic group and limiting long-standing illness.

**Results** Participants seeking social housing had the fewest daily steps (8304, 95% CI 7959 to 8648) and highest BMI (26.0 kg/m², 95% CI 25.5 kg/m² to 26.5 kg/m²) compared with those seeking intermediate (daily steps 9417, 95% CI 9106 to 9731; BMI 24.8 kg/m², 95% CI 24.4 kg/m² to 25.2 kg/m²) or market-rent housing (daily steps 9313, 95% CI 8858 to 9768; BMI 24.6 kg/m², 95% CI 24.0 kg/m² to 25.2 kg/m²). Those seeking social housing had lower levels of PA (by 19%–42%) at weekends versus weekdays, compared with other housing groups. Positive perceptions of neighbourhood quality were associated with higher steps and lower BMI, with differences between social and intermediate groups reduced by ~10% following adjustment, equivalent to a reduction of 111 for steps and 0.5 kg/m² for BMI.

**Conclusions** The social housing group undertook less PA than other housing sectors, with weekend PA offering the greatest scope for increasing PA and tackling adiposity in this group. Perceptions of neighbourhood quality were associated with PA and adiposity and reduced differences

### Strengths and limitations of this study

► Large sample with representation of three different aspirational housing groups, providing a wide range of socioeconomic backgrounds.
► Objective measurements of physical activity and adiposity outcomes using accelerometry and bioelectrical impedance respectively.
► Lower number of participants studied seeking market-rent housing compared with those seeking intermediate or social housing.

in steps and BMI between housing sectors. Interventions to encourage PA at weekends and improve neighbourhood quality, especially among the most disadvantaged, may provide scope to reduce inequalities in health behaviour.

## INTRODUCTION

Physical inactivity and adiposity are associated with an increased risk of type 2 diabetes and cardiovascular disease[1–4] and constitute a serious public health problem in the UK and globally.[5] Evidence suggests that levels of physical activity (PA) are lower among those who are socioeconomically disadvantaged,[6] who experience greater economic, access and health-related barriers to being physically active.[7] Socioeconomic status is also associated with differences in types of PA, in particular higher socioeconomic status is associated with more vigorous leisure time PA.[8] Previous research has found variation in PA by day of the week with studies showing lower levels of activity on Sundays compared with weekdays in young adults,[9] parents and their children.[10]

There is emerging evidence suggesting that housing tenure is an important determinant

of health. In particular, UK-based studies have shown that housing tenure (owner vs private renter vs public sector renter) is associated with poor health.[11 12] Among particular groups, including those who are economically inactive or unemployed, housing tenure might provide a better indication of socioeconomic status compared with measures based on occupation or income.[13] Indeed, in several studies, housing tenure remained associated with health outcomes following adjustment for conventional measures of socioeconomic status such as income or education.[11 14] A more nuanced approach is therefore required with respect to measures of socioeconomic status, and they should not be simply regarded as interchangeable.[12 15] Despite this, there has been limited research examining the direct effect of housing tenure on PA, and existing evidence is equivocal. Harrison and colleagues[16] found no association between housing tenure and meeting recommended levels of PA among community-dwelling healthy adults in the North-East of England. Similarly, housing tenure was not associated with self-reported energetic PA among older Australians.[17] Ogilvie and colleagues[18] found overall levels of PA to be higher among individuals living in social housing compared with owner–occupiers. The authors suggest that may capture occupational PA levels that are likely to be higher among those in social housing.[18] In contrast, living in private rental accommodation was associated with a greater likelihood of taking up exercise over a 9-year period among men aged 18–49 years at baseline, compared with those in local authority accommodation.[19]

Housing tenure may affect health and health behaviours in part through characteristics of the home or neighbourhood itself[20 21] or psychological factors such as self-efficacy or self-esteem.[22] Social housing estates that are common in the UK may be associated with specific cultures and norms, which in turn shape residents' behaviours.[11] Subjective characteristics of the neighbourhood environment including higher perceived access to recreational facilities and shops in local proximity have been shown to be associated with higher levels of PA.[23 24] Residents who perceive their neighbourhood more positively have been shown to have better mental health and are less likely to relocate.[25] Conversely, real and perceived crime, has the potential to constrain residents' PA.[26] However, a recent systematic review suggested a lack of association between PA and perceptions of safety from crime, highlighting the need for high-quality evidence, including prospective studies and natural experiments,[27] to examine this issue further. In particular, high-quality evidence is needed to understand the potentially multifactorial influence of residential location on health and health behaviours, effects that are likely to extend beyond simple measures of socioeconomic status.[27]

The Examining Neighbourhood Activities and Built Living Environments in London (ENABLE London) study is a longitudinal study evaluating how active urban design influences the health and well-being of people moving into the former Athletes' Village of the London 2012 Olympic and Paralympic Games now known as 'East Village'.[28] East Village is a new high-density neighbourhood development built on active design principles containing a mix of social housing, intermediate (including affordable rent, shared ownership and shared equity) housing and market-rent housing. This paper draws on baseline data (prior to any potential move to East Village) to first examine predictors of PA and adiposity (measured objectively using accelerometry and bioelectrical impedance), including the housing sector to which they are applying and perceptions of their neighbourhood. Second, to examine whether PA patterns across the week vary by housing sector, and third, to examine whether adjustment for perceptions of the neighbourhood environment reduce housing sector differences in PA and adiposity.

## METHODS

Study participants were recruited from those seeking or who had applied for new accommodation in East Village and were classified by the type of housing tenure sought based on level of income, that is, social, intermediate or market-rent. The inclusion criteria was broad and included anyone interested/applying for single or multiple occupancy accommodation in East Village. There was no explicit exclusion criteria; adults of any age, gender, ethnic group and with or without handicap were invited to participate. Current housing status was strongly linked to aspirational housing status, where those seeking social accommodation were currently in social housing or on social housing waiting lists, and those seeking intermediate and market-rent accommodation were largely in privately rented housing. Recruitment of participants in the different housing sectors was carried out between January 2013 and December 2015 in three phases determined by the order of availability of housing in East Village (social, intermediate and market-rent, respectively). Those applying for social housing in East Village were initially recruited between January 2013 and May 2014, households seeking intermediate accommodation between July 2013 and November 2014 and those seeking market rent accommodation between September 2014 and December 2015. Recruitment processes for those applying for social housing were slightly different compared with other housing sectors. The East Thames Group housing association was primarily responsible for recruiting participants in social housing, whereas the ENABLE London team (in association with Triathlon Homes and Get Living London) recruited participants from the other housing sectors.[28] Aspirational housing tenure is integral to the design of ENABLE London, and we have shown that this provides a clear socioeconomic marker of study participants. For example, those seeking social housing in East Village are more likely to be unemployed, less educated and more likely to represent ethnic minorities (a classic marker of socioeconomic vulnerability), compared with those seeking affordable

and market-rent accommodation.[28] We have also shown key differences in mental health and well-being between housing groups, where those seeking social housing were more likely to be depressed, anxious and have poorer well-being, compared with other housing groups.[29] Moreover, this is entirely consistent with earlier studies that found that both current housing tenure and aspirational housing tenure are associated with a variety of health outcomes, including mental health and measures of general health.[20 30]

Baseline assessments of participants were carried out in their place of residence before any potential move to East Village. Full details of the recruitment process can be found elsewhere.[28]

### Independent variables

A team of trained fieldworkers administered self-complete questionnaires on a laptop during home visits. Data on age, sex, self-defined ethnicity, work status, occupation and whether the participant had a limiting long-standing illness or disability (lasting or expected to last at least 12 months) were collected. Participants self-defined as 'White', 'Asian', 'Black', 'Mixed' or 'Other'; the latter two categories were combined for analyses.

Socioeconomic status based on occupation was coded using the National Statistics Social-Economic Coding to categorise participants into 'higher managerial or professional occupations', 'intermediate occupations' and 'routine or manual'.[31] An additional 'economically inactive' category included those seeking employment, unable to work due to disability or illness, retired, looking after home and family and students. We sought information on educational attainment; participants were categorised into 'Degree or equivalent/Higher', 'Intermediate qualifications' (including A levels and General Certificates of Secondary Education (GSCEs) and 'Other/None' (including work-based or foreign qualifications). Participants completed questionnaires assessing neighbourhood perceptions.[29] Five items assessed perceived crime (eg, 'There is a lot of crime in my neighbourhood'; Cronbach's α=0.87) and six items assessed neighbourhood quality (eg, 'This area is a place I enjoy living in'; Cronbach's α=0.78). Responses on items were summed and scores ranged from −10 to +10 for perceived crime and −12 to +12 for perceived quality, such that positive scores indicate less perceived crime and better neighbourhood quality, while negative scores indicate more perceived crime and poorer quality. The scales were derived following an exploratory factor analysis of 14 questions regarding neighbourhood (online supplementary table 1).

### Dependent variables

Height was measured to the last complete millimetre using a portable stadiometer; weight was measured to the nearest kilogram using a Tanita SC-240 Body Composition Analyzer (Tanita, Tokyo, Japan); body mass index (BMI) was derived as weight (kg)/height (m)$^2$. The Tanita SC-240 Body Composition Analyzer also measured leg-to-leg bioelectrical impedance from which fat free mass and fat mass were estimated. Fat mass percentage was calculated as fat mass (kg)/weight (kg)*100.

Participants wore a hip-mounted ActiGraph GT3X+ accelerometer during waking hours over a consecutive period of 7 days (ActiGraph LLC, Florida, USA). These accelerometers provided daily measures of steps, counts and time spent in moderate and vigorous PA (MVPA) using established cut-offs. Daily time spent in MVPA both overall and in ≥10 min bouts in accordance with UK recommendations for PA[32] were assessed. The cut-point for moderate PA was defined as ≥1952 counts per minute.[33] We excluded any days of recording where the amount of registered time accumulated was below 540 min.[34] Non-wear periods were defined as a minimum length of 60 min, allowing for a 2 min spike tolerance. Participants with at least 1 day of recording were retained in analyses. We fitted a multilevel linear model for each outcome to allow for repeated measurements of daily PA, by fitting participant as a random effect and adjusting for day of the week, day order of recording and month as fixed effects. Raw level one residuals were obtained from the model, and a within-person average value of each outcome variable was obtained by averaging these raw residuals. The average of these raw residuals for each participant was added to the sample mean for that particular PA variable to derive an unbiased average level of each PA variable for each person.

### Statistical analysis

All analyses were carried out using STATA/SE software (Stata/SE V.14 for Windows). Outcome variables were inspected for normality, and BMI was log transformed due to its skewed distribution. Multilevel linear regression models were fitted, mutually adjusted for housing sector and participant characteristics (sex, age group, ethnic group and limiting long-standing illness) as fixed effects, with a random effect to allow for household clustering. Residuals did not show departure from linearity, suggesting that the model assumptions were appropriate. Absolute differences or percentage differences for log transformed outcomes (ie, BMI) are presented by sex, age group, ethnic group, limiting long-standing illness and housing sector. Sensitivity analyses examined whether associations remained when the sample was restricted to 931 participants (84%) with at least 4 days of 540 or more minutes per day of recording.

To assess differences in PA by day of the week as opposed to overall levels of PA, we took the following approach. Daily PA data were examined using multilevel models with random effects to allow for multiple days of recording within person and household clustering. An interaction between housing sector and day of the week was fitted, and models were adjusted for sex, age group, ethnic group, limiting long-standing illness, day order of recording and month of measurement as fixed effects.

The associations between neighbourhood perception scales and adiposity and PA outcomes were examined. Each of the neighbourhood quality and crime scores were included in the models as quintiles to examine the differences in outcomes between the top and bottom quintile. Finally, the effect of adjustment for neighbourhood perception on differences in adiposity and PA between housing sectors was examined. If associations between outcomes and neighbourhood perceptions appeared linear, models examining housing sector differences were additionally adjusted for neighbourhood perceptions as a continuous variable.

### Patient and public involvement
The ENABLE London study was developed in partnership with a network of both local and regional stakeholders identified through our collaborator links to agencies, involved with the design, planning and management of large-scale accommodation developments. Locally, these included local authorities (particularly Newham) and a number of housing associations, in particular Triathlon Homes, a partner organisation of housing associations, which manages social and intermediate homes in East Village. Participants have been involved in the study from an early stage to ensure assessments and participation remain relevant and enjoyable to ensure the continued significance and potential generalisability of the work.

### RESULTS
Of 1819 households who agreed to be contacted by the study team in order to receive further information about the ENABLE London study, 1278 adults from 1006 households (55%) participated in the study and completed a questionnaire. Participation rates for those seeking market-rent and intermediate housing were 58% and 57%, respectively, and were slightly lower in the social group (52%). Complete data on adiposity were available for 1240 participants (97%); of these, a subset of 1107 participants (89%) met the inclusion criteria for analyses of objectively measured PA. Participant characteristics (age and sex) and levels of adiposity were similar among those who did and did not provide PA data; however, participants from black and Asian ethnic groups were less likely to provide PA data. Online supplementary table 2 shows participants characteristics at baseline for the 1240 adults with measurements of adiposity at baseline. Those seeking social housing were more likely to be female, of older age, of non-white ethnicity, to have limiting long-standing illness, and be in routine/manual occupations or economically inactive compared with those seeking intermediate or market-rent housing.

Adjusted mean levels of adiposity and PA outcomes by housing sector and participant characteristics are shown in online supplementary table 3. Table 1 shows housing sector and other participant characteristics associations with BMI and fat mass % and objectively measured PA (steps, time spent in MVPA and time spent in MVPA

in ≥10 min bouts). Participants seeking social housing had markedly higher levels of BMI and fat mass % and markedly lower levels of steps, MVPA and MVPA in ≥10 min bouts compared with those seeking intermediate housing, though there were no differences between those seeking market-rent and intermediate accommodation.

Fat mass % was higher in females than males though there was no difference in BMI (table 1). BMI and fat mass % were higher among all older age groups compared with 16–24 year olds. Participants of black ethnicity had higher levels of BMI and fat mass % compared with whites; there were no differences in BMI and fat mass % between Asian or other/mixed ethnic groups and whites. Those with a limiting long-standing illness had higher levels of both BMI and fat mass %. All PA measures were lower among females. Steps and MVPA were slightly higher in 25–34 year olds and steps were also higher among 35–49 year olds compared with 16–24 year olds; however, there were no age group differences for MVPA in ≥10 min bouts. Participants of black and Asian ethnicities had lower levels of steps, MVPA and MVPA in ≥10 min bouts compared with whites. Participants who reported having a limiting long-standing illness had lower levels of steps and MVPA but not MVPA in ≥10 min bouts. Educational attainment level was not associated with any of the outcomes once housing sector had been adjusted for, and adjustment for educational attainment did not materially alter housing sector differences in adiposity or PA outcomes (data available from authors).

Sensitivity analyses for PA outcomes were carried out in 931 participants who wore an ActiGraph for at least 4 days with at least 540 min of recording per day (online supplementary table 4). There were no differences between market-rent and intermediate groups (consistent with the main analysis presented in table 1). Differences between social and intermediate groups were broadly similar with the results presented in table 1 for the main analysis.

Differences in PA variables between housing groups were examined by day of the week to explore whether differences between groups were consistent across the week (figure 1A–D). Levels of PA (steps (panel A), MVPA (panel B) and MVPA in ≥10 min bouts (panel C)) were generally consistent across weekdays (Monday–Friday) among all groups. In the intermediate group, steps were higher on Saturdays and lower on Sundays; MVPA and MVPA in ≥10 min bouts were lower on Sundays, but there was no difference on Saturdays compared with weekday activity. In the market-rent group, steps, MVPA and MVPA in ≥10 min bouts were higher on Saturdays and similar to weekdays on Sundays. In the social group, steps, MVPA and MVPA in ≥10 min bouts were on average lower on Saturdays and lower still on Sundays. Registered time (panel D) was lowest on average in the social group during weekdays, decreasing on Saturdays and Sundays. The intermediate and market-rent groups had higher levels of registered time during weekdays compared with the social group that decreased on average on Saturdays and Sundays (despite recording more steps and minutes

**Table 1** Associations between adiposity and physical activity outcomes and patient characteristics

| | n | Difference or % difference* in adiposity/physical activity (95% CI), P values | | | | | | | | | |
| | | BMI (kg/m²)* | | Fat mass % | | Daily steps† | | Daily minutes spent in MVPA† | | Daily minutes spent in MVPA in ≥10 min bouts† | |
| **Sex** | | | | | | | | | | | |
| Male (Ref) | 522 | – | – | – | – | – | – | – | – | – | – |
| Female | 718 | −1.2 (−3.2 to 0.9) | 0.26 | 11.1 (10.3 to 12.0) | <0.0001 | −570 (−946 to −194) | 0.003 | −9.3 (−12.2 to −6.4) | <0.0001 | −4.1 (−6.1 to −2.0) | <0.001 |
| **Age group (years)** | | | | | | | | | | | |
| Age 16–24 (Ref) | 269 | – | – | – | – | – | – | – | – | – | – |
| Age 25–34 | 531 | 6.3 (3.5 to 9.1) | <0.0001 | 3.2 (2.1 to 4.3) | <0.0001 | 502 (11 to 992) | 0.04 | 4.0 (0.2 to 7.9) | 0.04 | 1.0 (−1.9 to 3.8) | 0.51 |
| Age 35–49 | 358 | 13.4 (10.2 to 16.6) | <0.0001 | 6.4 (5.2 to 7.6) | <0.0001 | 699 (173 to 1224) | 0.01 | 3.9 (−0.2 to 8.0) | 0.07 | −1.1 (−4.0 to 1.8) | 0.46 |
| Age 50+ | 82 | 17.6 (12.6 to 22.9) | <0.0001 | 9.2 (7.3 to 11.0) | <0.0001 | −9 (−832 to 813) | 0.98 | −6.0 (−12.4 to 0.5) | 0.07 | −2.0 (−6.8 to 2.7) | 0.40 |
| **Ethnic group** | | | | | | | | | | | |
| White (Ref) | 595 | – | – | – | – | – | – | – | – | – | – |
| Black | 314 | 6.2 (3.3 to 9.3) | <0.0001 | 3.6 (2.4 to 4.8) | <0.0001 | −1116 (−1657 to −575) | <0.0001 | −7.4 (−11.7 to −3.2) | <0.001 | −6.6 (−9.8 to −3.4) | <0.0001 |
| Asian | 210 | −0.3 (−3.1 to 2.7) | 0.85 | 0.02 (−1.2 to 1.3) | 0.97 | −1409 (−1972 to −845) | <0.0001 | −11.5 (−15.9 to −7.0) | <0.0001 | −8.1 (−11.4 to −4.8) | <0.0001 |
| Other/mixed | 121 | 1.3 (−2.3 to 5.0) | 0.48 | 1.0 (−0.5 to 2.5) | 0.18 | −430 (−1100 to 239) | 0.21 | −4.6 (−9.8 to 0.7) | 0.09 | −4.0 (−7.9 to −0.04) | 0.05 |
| **Limiting illness** | | | | | | | | | | | |
| No (Ref) | 1087 | – | – | – | – | – | – | – | – | – | – |
| Yes | 153 | 4.3 (1.1 to 7.5) | 0.01 | 1.6 (0.3 to 2.9) | 0.01 | −1081 (−1666 to −496) | <0.001 | −5.7 (−10.3 to −1.1) | 0.01 | −2.8 (−6.1 to 0.5) | 0.10 |
| **Housing sector** | | | | | | | | | | | |
| Social | 512 | 5.0 (2.2 to 7.8) | <0.001 | 2.7 (1.5 to 3.8) | <0.0001 | −1125 (−1629 to −620) | <0.0001 | −7.5 (−11.5 to −3.6) | <0.001 | −6.5 (−9.5 to −3.5) | <0.0001 |
| Intermediate (Ref) | 503 | – | – | – | – | – | – | – | – | – | – |
| Market-rent | 225 | −0.8 (−3.6 to 2.0) | 0.57 | −0.2 (−1.4 to 1.0) | 0.70 | −104 (−633 to 424) | 0.70 | 2.3 (−1.9 to 6.4) | 0.29 | 2.8 (−0.3 6.0) | 0.08 |

All differences and % differences are mutually adjusted for sex, age group, ethnic group, limiting long-standing illness, housing sector and a random effect to allow for clustering at household level.
MVPA and MVPA in ≥10 min bouts are an average daily estimate, obtained from averaging a participant's weekly total.
*Percentage differences are presented for BMI, which was log-transformed for analysis.
†Missing data for 133 participants.
BMI, body mass index; MVPA, moderate and vigorous physical activity.

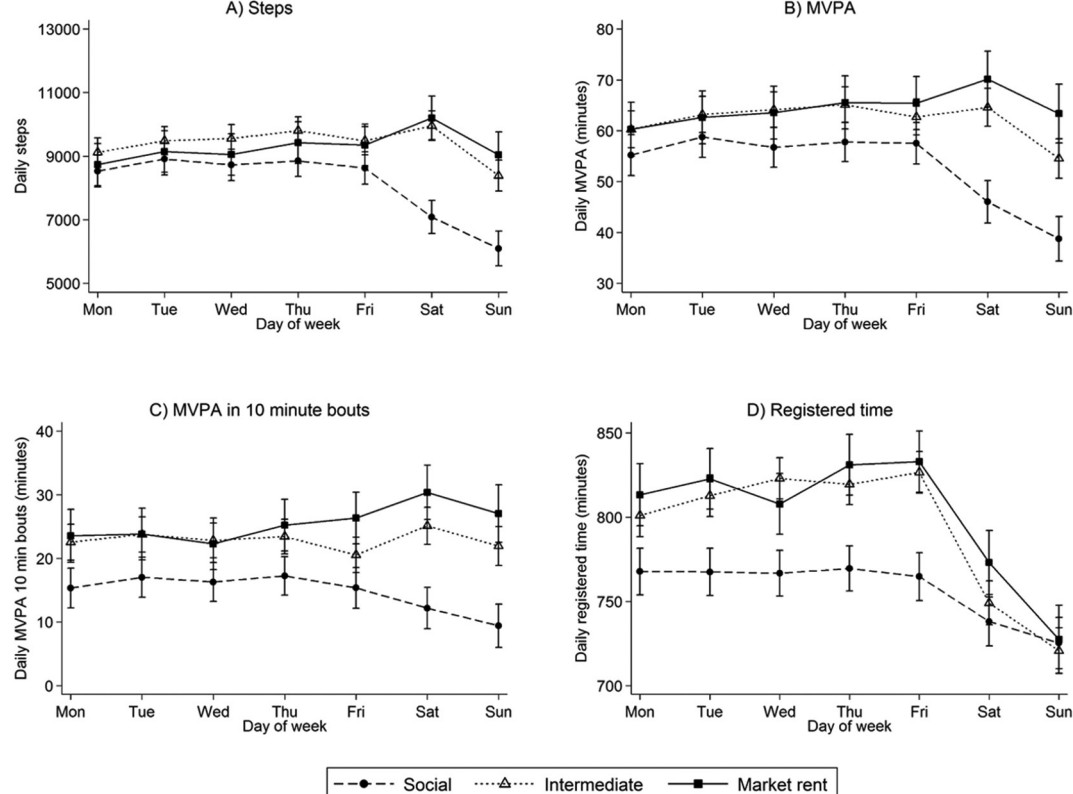

**Figure 1** Daily physical activity by day of the week and housing sector group: n=6206 days from 1107 participants. Means and 95% CI are adjusted for sex, age group, ethnic group, limiting long-standing illness, month of recording, day order of recording, day of week, housing sector, an interaction between housing sector and day of week and random effects to allow for multiple days of measurement and clustering of participants within households. MVPA, moderate and vigorous physical activity.

in MVPA suggesting a higher intensity of activity). Mean levels of steps, MVPA and MVPA in ≥10 min bouts on weekdays and differences on Saturday and Sunday compared with weekdays are shown by housing sector in online supplementary table 5. The marked differences in activity between weekdays and weekend days in the social group are not explained by differences in registered time (data available from authors).

Associations between perceived neighbourhood quality and crime scales and adiposity and PA outcomes are shown in table 2, adjusted for the participant characteristics shown in table 1. Participants with the most positive perceptions of neighbourhood quality (highest quintile) had lower BMI, higher steps and recorded longer durations of MVPA compared with those who had the most negative perceptions of neighbourhood quality (lowest

| **Table 2** Associations between adiposity and physical activity outcomes and neighbourhood perceptions scales | | | | |
|---|---|---|---|---|
| | **Difference or % difference\* in outcome between the highest and lowest quintiles for each neighbourhood scale (95% CI), p values** | | | |
| | **Perceptions of NH quality** | | **Perceptions of NH crime** | |
| Adiposity (n=1240) | | | | |
| Body mass index (kg/m$^2$)\* | −3.6 (−6.5 to −0.6) | 0.02 | −2.1 (−5.4 to 1.3) | 0.21 |
| Fat mass % | −1.2 (−2.5 to 0.06) | 0.06 | −0.8 (−2.2 to 0.7) | 0.30 |
| Physical activity (n=1107) | | | | |
| Daily steps | 677 (108 to 1247) | 0.02 | −63 (−713 to 587) | 0.85 |
| Daily MVPA (min) | 4.5 (0.02 to 9.0) | 0.05 | 1.1 (−4.0 to 6.2) | 0.68 |
| Daily MVPA in ≥10 min bouts (min) | 2.7 (−0.6 to 6.0) | 0.11 | 2.4 (−1.4 to 6.1) | 0.22 |

All differences and % differences are adjusted for sex, age group, ethnic group, limiting long-standing illness, housing sector and a random effect to allow for clustering at household level.
MVPA and MVPA in ≥10 min bouts are an average daily estimate, obtained from averaging a participant's weekly total.
\*Percentage differences are presented for BMI, which was log-transformed for analysis.
MVPA, moderate and vigorous physical activity; NH, neighbourhood.

**Table 3** Adiposity and physical activity differences between housing sectors: adjustment for perceptions of neighbourhood quality

| | Difference or % difference* compared with intermediate housing group (95% CI), p values | | | |
| --- | --- | --- | --- | --- |
| | **Model 1** | | **Model 2 (additionally adjusted for neighbourhood quality scale)** | |
| Adiposity  (n=1240) | | | | |
| Body mass index (kg/m$^2$)* | | | | |
| Social | 5.0 (2.2 to 7.8) | <0.001 | 4.5 (1.7 to 7.3) | 0.002 |
| Intermediate | Reference group | | | |
| Market rent | −0.8 (−3.6 to 2.0) | 0.57 | −0.9 (−3.6 to 2.0) | 0.55 |
| Fat mass % | | | | |
| Social | 2.7 (1.5 to 3.8) | <0.0001 | 2.5 (1.4 to 3.6) | <0.0001 |
| Intermediate | Reference group | | | |
| Market rent | −0.2 (−1.4 to 1.0) | 0.70 | −0.2 (−1.4 to 0.9) | 0.68 |
| Physical activity  (n=1107) | | | | |
| Daily steps | | | | |
| Social | −1125 (−1629 to −620) | <0.0001 | −1016 (−1531 to −501) | <0.001 |
| Intermediate | Reference group | | | |
| Market rent | −104 (−633 to 424) | 0.70 | −96 (−624 to 431) | 0.72 |
| Daily MVPA (min) | | | | |
| Social | −7.5 (−11.5 to −3.6) | <0.001 | −6.8 (−10.8 to −2.7) | 0.001 |
| Intermediate | Reference group | | | |
| Market rent | 2.3 (−1.9 to 6.4) | 0.29 | 2.3 (−1.8 to 6.5) | 0.27 |
| Daily MVPA in ≥10 min bouts (min) | | | | |
| Social | −6.5 (−9.5 to −3.5) | <0.0001 | −6.0 (−9.1 to −3.0) | <0.001 |
| Intermediate | Reference group | | | |
| Market rent | 2.8 (−0.3 to 6.0) | 0.08 | 2.8 (−0.3 to 6.0) | 0.08 |

Model 1: adjusted for sex, age group, ethnic group, limiting longstanding illness and clustering at household level (random effect).
Model 2: adjusted as Model 1 plus neighbourhood quality scale (added as a continuous variable).
MVPA and MVPA in ≥10 min bouts are an average daily estimate, obtained from averaging a participant's weekly total.
*Percentage differences are presented for BMI, which was log-transformed for analysis.
BMI, body mass index; MVPA, moderate and vigorous physical activity.

quintile). There were no significant associations between perceptions of neighbourhood crime and adiposity or PA.

The effect of adjustment for perceived neighbourhood quality on differences in adiposity and PA between housing sector groups is presented in table 3. All associations between perceived neighbourhood quality and crime, and outcome variables were approximately linear and were therefore fitted as continuous variables in the model. In addition, associations between perceived neighbourhood quality and crime and outcome variables were similar across the three housing groups (all p>0.05). Adjustment for perceptions of neighbourhood quality reduced differences in BMI, fat mass %, steps, MVPA and MVPA in ≥10 min bouts between the social and intermediate groups by 10%, 6%, 10%, 10% and 7%, respectively. Differences between market-rent and intermediate groups in adiposity and PA variables were not statistically significant before or after adjustment. A larger proportion of the social-intermediate group differences in steps, MVPA

and MVPA in ≥10 min bouts on weekends was explained by adjustment for perceptions of neighbourhood quality (10%, 16% and 16%, respectively) compared with the differences in steps, MVPA and MVPA in ≥10 min bouts on weekdays, which were reduced by 10%, 8% and 3%, respectively (data not shown).

## DISCUSSION

The results of this study showed that participants seeking social housing in East Village had lower levels of PA and higher levels of BMI and fat mass % compared with those seeking intermediate and market-rent housing, even when adjusted for demographic factors. In the social housing group, levels of PA were particularly low on weekends compared with weekdays possibly reflecting higher occupational PA and lower leisure time PA; weekday–weekend differences in PA were less marked among those seeking intermediate and market-rent housing. However,

the lower registered time at weekends but higher MVPA and steps suggests more intense activity at weekends in the intermediate and market-rent housing groups. These findings may inform targeted interventions to increase PA and reduce adiposity in different socioeconomic groups.

Positive associations between perceived neighbourhood quality and PA, BMI and fat mass % were also shown. Adjustment for differences in perceived neighbourhood quality reduced differences in PA and BMI by approximately 10% between social and intermediate housing groups, equivalent to a reduction of 111 for daily steps, 0.5 min for MVPA and 0.5 kg/m² for BMI. However, a larger proportion of the difference in PA was apparent at weekends, equivalent to a reduction of 222 for daily steps and 2.2 min for MVPA.

### Relation to previous studies

Studies have shown that lower socioeconomic status is associated with lower levels of PA[35 36] and that those from more socially deprived backgrounds have the most barriers to being physically active.[7] Previous research examining the role of housing tenure is limited. Findings from this study showed marked differences in PA and adiposity between those seeking social, intermediate and market-rent housing. In particular, lower PA and higher adiposity in participants seeking social housing, a group that comprises a high proportion of people from more socioeconomically disadvantaged backgrounds.[28] The higher levels of BMI and fat mass % in those seeking social housing compared with those seeking intermediate or market-rent housing is consistent with systematic reviews that have found an association between lower socioeconomic status and higher levels of adiposity, particularly in higher income countries and among women.[37] While socioeconomic status is a strong determinant of housing status, to our knowledge this is the first study to explicitly examine housing sector differences in objective PA and markers of adiposity levels (ie, BMI and fat mass %). However, it is important to consider more broadly what these aspirational housing sector differences might represent. Related studies have shown that those in social housing are less likely to use active travel compared with owner occupiers[18] and that those in social housing and home owners with a mortgage are more likely to be obese and have higher levels of illness and disability compared with outright home owners, even after adjustment for other socioeconomic status markers.[38] These latter findings suggest that the effect of home ownership may be more complex and cannot be simply explained by socioeconomic status. Neighbourhood quality may offer a potential partial explanation for these findings.[39] In the present study, perceptions of better neighbourhood quality were associated with PA, whereas perceptions of crime were not. In contrast, a large UK-based study found that perceptions of feeling safe in the neighbourhood had the largest effect on levels of PA compared with perceptions of leisure facilities, sense of belonging or access to public transport or amenities.[40] Another study in the USA found that low perceived safety from crime was associated with lower levels of MVPA.[41] However, a recent review concluded that higher quality evidence is needed, including prospective studies and natural experiments in areas of wide crime variability, in order to further understand the effect of crime on physical and mental health.[27] Moreover, previous work has suggested that objective and perceived measures of the built environment correlate differently with PA levels, suggesting that these measures are assessing different dimensions of the built environment, which relate differently to health behaviour.[42]

Our findings showed that PA levels were particularly low on the weekend among those seeking social housing, which is consistent with findings from a systematic review that found that leisure time PA (which may be more likely to occur on weekends) was lower among those from lower socioeconomic groups.[8] This suggests that low-cost strategies to increase weekend PA may be particularly beneficial to more disadvantaged households. A free community-based programme in Bogata Colombia temporarily closed streets on Sundays to encourage PA among more disadvantaged local residents.[43] A similar programme has been trialled in the USA[44]; however, the effectiveness, longevity and generalisability of these programmes to other socioeconomically deprived areas is yet to be established.

### Strengths and limitations

Strengths of this study include the representation of three different aspirational housing groups that provides a wide range of socioeconomic backgrounds. Of those seeking social housing, two-thirds (67%) were currently living in social housing accommodation provided by the local authority or housing association; the remainder were largely currently living in privately rented accommodation with many on social housing waiting lists. Of those seeking intermediate or market-rent accommodation, almost two-thirds were living in privately rented accommodation (both 64%); the remainder were largely living with relatives or friends. The study sample is large with good representation from a 'hard to reach' group of social housing participants. Participation rates were high given the target group, with between 50% and 60% of those who initially agreed to be contacted taking part in the study. The ActiGraph GT3X+ accelerometer provided validated objective measures of PA[45] and the use of bioelectrical impedance to provide more direct measurements of adiposity including fat mass %, which may provide a more valid marker of adiposity than BMI, particularly in a multi-ethnic population.[46 47] Reassuringly, the patterns of PA by sex, ethnic group and health status were consistent with those published previously.[48–50] A limitation of the study is the lower number of participants in the market-rent sector compared with the other groups. This was due to restrictions imposed on the study team on the extent and duration of access to potential applicants seeking market-rent accommodation. While the study is longitudinal, these analyses are cross-sectional, limiting the degree to

which causal inferences can be made. Moreover, there is the possibility of selection among study participants, where those who are more active seek to move to East Village, may be more likely to participate in the study and may perceive their environment differently, which may limit the generalisability of the findings to neighbourhoods outside of East London.

## Conclusions and future work

The findings presented in this paper suggest that perceived neighbourhood quality is associated with meaningful differences in PA and markers of adiposity. Differences in steps (680 steps) and BMI ($3.6 \text{kg/m}^2$) between the lowest and highest quintiles of perceived neighbourhood quality should be considered in the context of an average 10 000 steps per day, where a 5% increase (500 steps) would be a worthwhile population level increase and a $5 \text{kg/m}^2$ increase in BMI is associated with a 31% increase in all-cause mortality.[51] Hence, improvements in neighbourhood quality could be associated with health benefits of public health importance. There were also substantial differences in PA, BMI and fat mass % between the three housing groups studied. In particular, the very low levels of PA in the social housing group during the weekend could provide a target for intervention to increase levels of PA; again, these differences should be considered in relation to 500 steps per day, which can be considered as an increase of population importance. Perceptions of neighbourhood quality reduced differences in PA and adiposity between housing sector groups, and the possibility of measuring more objective markers of neighbourhood quality within this study has the potential to explain more.[42] The future follow-up of the ENABLE London cohort will allow us to examine whether moving to 'East Village', a neighbourhood designed for healthy active living, will have a positive impact on PA and/or adiposity levels. A major aim of the study is to identify features of the local built environment that increase levels of PA that could potentially help to reduce socioeconomic inequalities in health. It will be of particular interest to determine whether an increase in PA is more apparent in the social housing group whose neighbourhood characteristics should improve. Furthermore, we will be in a position to examine whether any potential effects of the built environment on PA are modified by housing sector type.

## Author affiliations

[1]Population Health Research Institute, St George's University of London, London, UK
[2]Centre for Exercise, Nutrition and Health Sciences, University of Bristol, Bristol, UK
[3]National Institute for Health Research Bristol Biomedical Research Centre, University Hospitals Bristol NHS Foundation Trust and University of Bristol, Bristol, UK
[4]MRC/SCO Social and Public Health Sciences Unit, University of Glasgow, Glasgow, UK
[5]NHMRC Centre of Research Excellence in Healthy Liveable Communities, RMIT University, Melbourne, Victoria, Australia
[6]Department of Public Health, Environments and Society, London School of Hygiene and Tropical Medicine, London, UK

**Acknowledgements** The authors would like to thank the East Thames Group, Triathlon Homes and Get Living London who have assisted in recruiting participants into the ENABLE London study. The ENABLE London study is advised by a Steering Committee chaired by Professor Hazel Inskip (University of Southampton), with Dr David Ogilvie (University of Cambridge) and Professor Andy Jones (University of East Anglia) as academic advisors and Mrs Kate Worley (formerly East Thames Group Assistant Director for Strategic Housing) as the lay/stakeholder member. The authors are grateful to the members of the ENABLE London study team (in particular Aine Hogan, Katrin Peuker, Cathy McKay) and to participating households, without whom this study would not be possible.

**Contributors** CGO, ARR, AE, ARC, DL, SC, BG-C, DGC and PHW designed the study and raised funding. BR, ARR, CC, DP and CGO collected data for the study; BR, ARR and CGO enrolled participants. CMN, BR, ESL, ARR, CC, DP and CGO undertook data management. CMN and ESL analysed the data; CMN wrote the first draft of the report. ARR, BR, ESL, AS, DP, ARC, ASP, AE, BG-C, CC, DL, SC, PHW, DGC and CGO critically appraised the manuscript and approved the final draft. CGO is responsible for data integrity.

**Funding** This research is being supported by project grants from the Medical Research Council National Prevention Research Initiative (MR/J000345/1) and National Institute for Health Research (NIHR) (12/211/69). Diabetes and obesity prevention research at St George's, University of London, is supported by the NIHR Collaboration for Leadership in Applied Health Research and Care (CLAHRC) South London. AE is funded by the UK Medical Research Council as part of the Neighbourhoods and Communities Programme (MC_UU_12017-10). BG-C is supported by a National Health and Medical Research Council (NHMRC) Principal Research Fellowship (#1107672). ARC and ASP are supported by NIHR Biomedical Research Centre at University Hospitals Bristol NHS Foundation Trust and the University of Bristol.

**Disclaimer** The views expressed in this publication are those of the author(s) and not necessarily those of the NHS, the National Institute for Health Research or the Department of Health.

**Competing interests** None declared.

**Patient consent** Not required.

**Ethics approval** Full ethical approval was obtained from the relevant Multi-Centre Research Ethics Committee (REC Reference 12/LO/1031). All participants provided written informed consent.

**Provenance and peer review** Not commissioned; externally peer reviewed.

**Data sharing statement** Further details of the ENABLE London study are available from the study website (http://www.enable.sgul.ac.uk/). We welcome proposals for collaborative projects. For general data sharing inquiries, contact Professor Owen ( cowen@sgul.ac.uk).

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
