## [Reviewer comments · BMJ Open]

ARTICLE DETAILS

TITLE (PROVISIONAL)	Housing, neighbourhood and sociodemographic associations with adult levels of physical activity and adiposity: baseline findings from the ENABLE London Study
AUTHORS	Owen, C; Nightingale, C; Rudnicka, Alicja; Ram, Bina; Shankar, Aparna; Limb, Elizabeth; Procter, Duncan; Cooper, Ashley; Page, Angie; Ellaway, Anne; Giles-Corti, Billie; Clary, Christelle; Lewis, Daniel; Cummins, Steven; Whincup, Peter; Cook, Derek

VERSION 1 – REVIEW

REVIEWER	Miranda Pallan University of Birmingham, UK
REVIEW RETURNED	17-Mar-2018

GENERAL COMMENTS	This is a very well written manuscript, which communicates the design and findings of a complex cross-sectional study clearly. The ENABLE London study, is an interesting natural experiment examining the effect of housing and surrounding environment on physical activity and health. It is good to see some preliminary data from this important study. I have a few minor suggestions which would give readers further information and also clarify parts of the manuscript: 1. Abstract – The last sentence of the abstract concludes that moving to East Village may provide scope to encourage PA and reduce adiposity. Whilst in the main manuscript, the authors describe their future work to explore this, in the abstract the sentence does not seem to follow. I would recommend just making conclusions on the findings of this study in the abstract.2. Aims and methods – it is not quite clear to me why the participants are classified on the accommodation that they are seeking to move to. The authors mention in the discussion that the majority of those seeking to move to the three types of housing in East Village are already living in a similar housing type, but I feel some more up front explanation in the methods of why the participants are classified in this way for comparison would be beneficial (rather than having the sampling frame as all those seeking to move to East Village and then classifying them on their current housing type).3. Methods – Although the recruitment process is described elsewhere, I would recommend including a little more information on participant recruitment, inclusion criteria, recruiting members of the same family etc.
--

	4. Statistical analysis section, line 26 - there seems to be a missing word: "Absolute differences of percentage differences for [missing word] are presented by...." 5. Results – participant recruitment: the authors mention how many consented to initial contact by the study team, can they report how many were initially contacted for this consent? This would help readers assess representativeness. 6. Results – Page 12 lines 46 onwards, and page 13 lines 1-20: This may be my misunderstanding but I am not quite clear on what analysis was done and what is presented in Table 2. My understanding is that scores for perceptions of neighbourhood quality and crime have been included in the models as continuous variables, but then in Table 2 the difference between highest and lowest quintile neighbourhood scores has been presented. It would be good for the authors to clarify exactly what they have done, as it seems they have done one type of analysis but presented another. 7. Discussion page 15 lines 51-54 – I would recommend rewording the following sentence as I found this a little confusing: "Moreover, effects of perceived and objective measures of neighbourhood quality may have differing and potentially independent effects on health behaviours including PA".
--	--

REVIEWER	Katelyn Holliday University of North Carolina, USA
REVIEW RETURNED	18-Apr-2018

GENERAL COMMENTS	This study reports baseline data for a natural experiment examining physical activity and adiposity within individuals interested in moving to a community designed for active living. The sample includes a mix of individuals interested in social (public) housing, intermediate (affordable/shared rent), and market-rent housing. Minor Comments The introduction uses several different terms for social housing. It may be helpful to define towards the beginning and use the same term throughout. Line 16-19 of page 9: What was the distribution for PA variables? Were there a lot of zeros that would make a different (zero-inflated, negative binomial, etc.) modeling strategy more appropriate? Line 26-29 of page 9: Is there a word missing? "Absolute differences or percentage differences for [] are presented by..." The results, discussion, and tables use a mix of the terms BMI, fat mass, adiposity, and body size. It would be helpful to choose one word for each outcome and use it consistently. Supplementary Table 3 (referenced in Line 45 of page 10) has adjusted mean levels of the PA outcomes. Are these for the entire monitoring period or average per day? Would be helpful to clarify in title. What are the values in the parentheses in the tables (95% CI, IQR, etc)? They aren't labeled in all cases.
---

	Lines 21-24 of page 11 and Table 1: Given that PA tends to increase with age, do you have thoughts on why the 16-24 year olds had fewer steps than the 25-49 year olds? Throughout the results and discussion, it would be helpful to comment on which of the statistically significant differences in adiposity and PA levels you think are meaningfully different values (e.g. do you view the association with positive perceptions of neighborhood reported in the last abstract results sentence to be meaningful? Why or why not?). Line 56 of page 17. I believe the word “of” is missing “A major aim of the study is to identify features of the local built environment that increase levels [of] PA...” Table 1: The “No” category N for “limiting illness” is on 2 lines Several tables have “Difference/% difference in...” It might be less confusing to have “Difference or % Difference” Figure 1. Might be helpful to use a different marker type for the 3 housing categories so that those not printing in color can easily determine the groups.
--	--

VERSION 1 – AUTHOR RESPONSE

Reviewer 1 – Dr Miranda Pallan

This is a very well written manuscript, which communicates the design and findings of a complex cross-sectional study clearly. The ENABLE London study, is an interesting natural experiment examining the effect of housing and surrounding environment on physical activity and health. It is good to see some preliminary data from this important study.

Author’s response: We thank the Reviewer for these favourable comments about our work.

1.1. Abstract – The last sentence of the abstract concludes that moving to East Village may provide scope to encourage PA and reduce adiposity. Whilst in the main manuscript, the authors describe their future work to explore this, in the abstract the sentence does not seem to follow. I would recommend just making conclusions on the findings of this study in the abstract.

Author’s response: We thank the Reviewer for this comment.

Changes to the paper: The last sentence of the Abstract has been changed to highlight that interventions to encourage physical activity at weekends and improve neighbourhood quality, especially amongst the most disadvantaged, may provide scope to reduce inequalities in health behaviour.

1.2. Aims and methods – it is not quite clear to me why the participants are classified on the accommodation that they are seeking to move to. The authors mention in the discussion that the majority of those seeking to move to the three types of housing in East Village are already living in a similar housing type, but I feel some more up front explanation in the methods of why the participants are classified in this way for comparison would be beneficial (rather than having the sampling frame as all those seeking to move to East Village and then classifying them on their current housing type).

Author's response: We agree that further detail about the use of aspirational housing sector as a key socioeconomic marker would be helpful.

Changes to the paper: We have expanded the text in the 'Methods' considerably, to provide further details about rationale for using aspirational housing status as a key exposure variable. Aspirational housing tenure is integral to the design of ENABLE London, and we have shown that this provides a clear socioeconomic marker of study participants. For example, those seeking social housing in East Village are more likely to be unemployed, less educated and more likely to represent ethnic minorities (a classic marker of socioeconomic vulnerability), compared to those seeking affordable and market-rent accommodation.¹ We have also shown key differences in mental health and well-being between housing groups, where those seeking social housing were more likely to be depressed, anxious and have poorer well-being, compared to other housing groups.² This is entirely consistent with earlier studies which found that both current housing tenure and aspirational housing tenure are associated with a variety of health outcomes, including mental health and measures of general health.^{3,4} Moreover, as the reviewer correctly recognises we observe no differences in associations with physical activity and adiposity outcomes, when using current as opposed to aspirational housing status.

1.3. Statistical analysis section, line 26 - there seems to be a missing word: "Absolute differences of percentage differences for [missing word] are presented by...."

Author's response: We thank the Reviewer for pointing this out. The missing word has been added.

Changes to the paper: The sentence has been changed to read 'Absolute differences or percentage differences for log transformed outcomes (i.e., BMI), are presented by sex, age group, ethnic group, limiting longstanding illness and housing sector'.

1.5. Results – participant recruitment: the authors mention how many consented to initial contact by the study team, can they report how many were initially contacted for this consent? This would help readers assess representativeness.

Author's response: Unfortunately it is not possible to give a precise denominator as the number approached about the study is unknown, as different personnel were involved in the recruitment strategies for different housing groups. The number provided gives the number who were given a leaflet about the study and agreed to be contacted to be provided with further information.

Changes to the paper: We have altered the text to make this clear, that the participation rate represents those who agreed to take part from those who expressed interest in the study and agreed to receive further information about the study.

1.6. Results – Page 12 lines 46 onwards, and page 13 lines 1-20: This may be my misunderstanding but I am not quite clear on what analysis was done and what is presented in Table 2. My understanding is that scores for perceptions of neighbourhood quality and crime have been included in the models as continuous variables, but then in Table 2 the difference between highest and lowest quintile neighbourhood scores has been presented. It would be good for the authors to clarify exactly what they have done, as it seems they have done one type of analysis but presented another.

Author's response: We thank the reviewer for bringing this issue to our attention. The results in Table 2 are from regression models where neighbourhood quality and crime scores were fitted as a 5-level categorical variable representing the quintiles, but this was not made clear in the methods section.

Changes to the paper: We have edited the Methods, Results and footnote to Table 2 to indicate that regression coefficients compare the lowest with the highest quintile of perception scores, as opposed to expressing the coefficient for a unit difference in perceptions scores.

1.7. Discussion page 15 lines 51-54 – I would recommend rewording the following sentence as I found this a little confusing: "Moreover, effects of perceived and objective measures of neighbourhood quality may have differing and potentially independent effects on health behaviours including PA."

Author's response: Agreed – the sentence has been changed to improve the readability.

Changes to the paper: The sentence now read 'Moreover, previous work has suggested that objective and perceived measures of the built environment correlate differently with physical activity levels, suggesting that these measures are assessing different dimensions of the built environment which relate differently to health behaviour ⁵.'

Reviewer 2 – Dr Katelyn Holliday

2.1. The introduction uses several different terms for social housing. It may be helpful to define towards the beginning and use the same term throughout.

Author's response: We thank the Reviewer for raising this issue.

Changes to the paper: We have homogenised 'social housing' terminology throughout.

2.2. Line 16-19 of page 9: What was the distribution for PA variables? Were there a lot of zeros that would make a different (zero-inflated, negative binomial, etc.) modeling strategy more appropriate?

Author's response: We can reassure the reviewer that recording zero counts while wearing ActiGraph accelerometers is uncommon, and non-wear periods were defined to minimise zero recordings when the devices are not being worn. In terms of the distribution of physical activity outcomes, steps (the main outcome) and total MVPA showed a near normal distribution, but more importantly, examination of residuals did not show departure from linearity, suggesting that the model assumptions were appropriate.

Changes to the paper: We have added a comment to the 'Statistical analysis' section to reassure the reader that residuals from the multilevel linear regression models were examined and did not show departure from linearity, suggesting that the model assumptions were appropriate.

2.3. Line 26-29 of page 9: Is there a word missing? "Absolute differences or percentage differences for [] are presented by..."

Author's response: We thank the reviewer for spotting this omission. We have corrected the sentence by adding 'log transformed outcomes (i.e., BMI)' accordingly.

Changes to the paper: We have amended the sentence to read 'Absolute differences or percentage differences for log transformed outcomes (i.e., BMI), are presented by sex, age group, ethnic group, limiting longstanding illness and housing sector'. See response to 1.3 above.

2.4. The results, discussion, and tables use a mix of the terms BMI, fat mass, adiposity, and body size. It would be helpful to choose one word for each outcome and use it consistently.

Author's response: Agreed.

Changes to the paper: We have revised the results, discussion and tables to homogenise the terminology used to describe measures of adiposity, i.e., use of BMI and fat mass percentage.

2.5. Supplementary Table 3 (referenced in Line 45 of page 10) has adjusted mean levels of the PA outcomes. Are these for the entire monitoring period or average per day? Would be helpful to clarify in title.

Author's response: For each PA outcome, an adjusted daily average was obtained for each participant as described in the Methods section. The data presented are means of these daily averages mutually adjusted for other participant characteristics.

Changes to the paper: We have altered the column headings in Table 3 to be clearer.

2.6. What are the values in the parentheses in the tables (95% CI, IQR, etc)? They aren't labeled in all cases.

Author's response: We thank the Reviewer for pointing out this inconsistency.

Changes to the paper: Labels have been added to the headers and / or footnotes of tables throughout, to make clear the data provided in parenthesis.

2.7. Throughout the results and discussion, it would be helpful to comment on which of the statistically significant differences in adiposity and PA levels you think are meaningfully different values (e.g. do you view the association with positive perceptions of neighborhood reported in the last abstract results sentence to be meaningful? Why or why not?).

Author's response: We thank the Reviewer for raising this issue. We agree that considering the public health importance of differences in physical activity and adiposity between exposure groups would be informative.

Changes to the paper: We have added the following text to the 'Conclusion and future work' section of the Discussion to put the findings in context 'The findings presented in this paper suggest that perceived neighbourhood quality is associated with meaningful differences in PA and markers of adiposity. Differences in steps (680 steps) and BMI (3.6kg/m²) between the lowest and highest quintiles of perceived neighbourhood quality should be considered in the context of an average 10,000 steps per day, where a 5% increase (500 steps) would be a worthwhile population level increase and a 5kg/m² increase in BMI is associated with a 31% increase in all-cause mortality (53). Hence, improvements in neighbourhood quality could be associated with health benefits of public health importance. There were also substantial differences in PA, BMI and fat mass % between the three housing groups studied. In particular, the very low levels of PA in the social housing group during the weekend could provide a target for intervention to increase levels of PA; again these differences should be considered in relation to 500 steps per day, which can be considered as an increase of population importance.'

2.8. Line 56 of page 17. I believe the word "of" is missing "A major aim of the study is to identify features of the local built environment that increase levels [of] PA..."

Author's response: We thank the Reviewer for spotting this omission.

Changes to the paper: The sentence has been updated accordingly.

2.9. Table 1: The "No" category N for "limiting illness" is on 2 lines.

Changes to the paper: Table 1 has been updated accordingly.

2.10. Several tables have "Difference/% difference in..." It might be less confusing to have "Difference or % Difference"

Changes to the paper: The Tables indicating 'difference/% difference' have been updated to 'difference or % difference'.

2.11. Figure 1. Might be helpful to use a different marker type for the 3 housing categories so that those not printing in color can easily determine the groups.

Author's response: We thank the Reviewer for this suggestion.

Changes to the paper: We have amended Figure 1 to be in black and white and with different markers: social housing group solid circles dashed line, intermediate open triangles dotted line and market rent solid squares and solid line.

Supporting references:-

- (1) Ram B, Nightingale CM, Hudda MT, Kapetanakis VV, Ellaway A, Cooper AR et al. Cohort profile: Examining Neighbourhood Activities in Built Living Environments in London: the ENABLE London-Olympic Park cohort. *BMJ Open* 2016; 6(10):e012643.
- (2) Ram B, Shankar A, Nightingale CM, Giles-Corti B, Ellaway A, Cooper AR et al. Comparisons of depression, anxiety, well-being, and perceptions of the built environment amongst adults seeking social, intermediate and market-rent accommodation in the former London Olympic Athletes' Village. *Health Place* 2017; 48:31-39.
- (3) Macintyre S, Ellaway A, Hiscock R, Kearns A, Der G, McKay L. What features of the home and the area might help to explain observed relationships between housing tenure and health? Evidence from the west of Scotland. *Health Place* 2003; 9(3):207-218.
- (4) Mason KE, Baker E, Blakely T, Bentley RJ. Housing affordability and mental health: does the relationship differ for renters and home purchasers? *Soc Sci Med* 2013; 94:91-97.

- (5) McGinn AP, Evenson KR, Herring AH, Huston SL, Rodriguez DA. Exploring associations between physical activity and perceived and objective measures of the built environment. *J Urban Health* 2007; 84(2):162-184.
- (6) Global BMI MC, Di AE, Bhupathiraju S, Wormser D, Gao P, Kaptoge S et al. Body-mass index and all-cause mortality: individual-participant-data meta-analysis of 239 prospective studies in four continents. *Lancet* 2016; 388(10046):776-786.

VERSION 2 – REVIEW

REVIEWER	Miranda Pallan University of Birmingham, UK
REVIEW RETURNED	04-Jun-2018

GENERAL COMMENTS	The authors have comprehensively addressed my previous comments.
--

REVIEWER	Katelyn Holliday University of North Carolina
REVIEW RETURNED	11-Jun-2018

GENERAL COMMENTS	The authors have adequately addressed the initial comments I provided.
--